# Correlations between behavior and hormone concentrations or gut microbiome imply that domestic cats (*Felis silvestris catus*) living in a group are not like 'groupmates'

**Hikari Koyasu** *, **Hironobu Takahashi, Moeka Yoneda, Syunpei Naba, Natsumi Sakawa, Ikuto Sasao, Miho Nagasawa, Takefumi Kikusui**

Laboratory of Human-Animal Interaction and Reciprocity, Azabu University, Kanagawa, Japan

* h.koyasu.k@carazabu.com

**Data Availability Statement:** All relevant data are within the manuscript and its Supporting Information files. The genetic data were registered

## Abstract

Domestic cats (*Felis silvestris catus*) can live in high densities, although most feline species are solitary and exclusively territorial animals; it is possible that certain behavioral strategies enable this phenomenon. These behaviors are regulated by hormones and the gut microbiome, which, in turn, is influenced by domestication. Therefore, we investigated the relationships between the sociality, hormone concentrations, and gut microbiome of domestic cats by conducting three sets of experiments for each group of five cats and analyzing their behavior, hormone concentrations (cortisol, oxytocin, and testosterone), and their gut microbiomes. We observed that individuals with high cortisol and testosterone concentrations established less contact with others, and individuals with high oxytocin concentrations did not exhibit affiliative behaviors as much as expected. Additionally, the higher the frequency of contact among the individuals, the greater the similarity in gut microbiome; gut microbial composition was also related to behavioral patterns and cortisol secretion. Notably, individuals with low cortisol and testosterone concentrations were highly tolerant, making high-density living easy. Oxytocin usually functions in an affiliative manner within groups, but our results suggest that even if typically solitary and territorial animals live in high densities, their oxytocin functions are opposite to those of typically group-living animals.

## Introduction

During evolution, ecological factors, including food availability, influence the development of group and solitary living behavior in animals [1]. Concentrated food distribution has led to selection for communal feeding, which has resulted in cooperative sociality [1]. Moreover, such sociality could have been enhanced by human interventions that affected ecological conditions, such as availability of new food resources in garbage dumps and destruction of pre-existing habitats [2]. Although most wild felids live alone [3], domesticated cats (*Felis silvestris catus*) live in high densities and interact with each other [4]. Incidentally, as humans began to

DNA Data Bank of Japan (https://www.ddbj.nig.ac.jp/index.html: accession number PRJDB12856).

**Funding:** This work was supported by the Japan Society for the Promotion of Science and Grants-in-Aid for Scientific Research from the Ministry of Education, Culture, Sports, Science, and Technology of Japan (Grant numbers: #20J14760 to H.K., 23 #18H02489 and #19K22823 to M.N., and #19H00972 to T.K.). The funders had no role in study design, data collection and analysis, decision to publish, or preparation of the manuscript.

**Competing interests:** The authors have declared that no competing interests exist.

settle down and start cultivation, cats acquired a new niche where food resources, such as rodents, were abundant, which led to an ability to live in high densities. In this process of acquiring a new niche, temperamental selection for communal feeding must have occurred in the felids, although not to an extent similar to that in the canids [5].

When solitary and exclusively territorial animals begin to live in groups due to domestication, it is natural to wonder whether their group-formation strategies are similar to those of typical group-living animals. They may be 'groupmates,' similar to other mammals that live in groups, or they may be in conflict with each other, in spite of living in high densities, due to their inherent solitary nature. Therefore, investigating how cats establish groups (compared to the group formation of other animal species) can help clarify the effects of ecological environment or nature of species on the process of group formation. Moreover, if cats live as 'groupmates,' then it would be possible to understand the mechanisms of animal group formation by exploring the factors influencing the development of 'groupmates' among cats.

Previously, the social behavior of animals living in groups has been studied on the basis of endocrine activity, such as glucocorticoids (GCs), testosterone, and oxytocin [6–17]. The primary physiological function of GCs is to increase the metabolism of glucose (energy) as per the requirement of behavioral responses. This energy production is necessary when an animal is faced with some threat, and it has been demonstrated that individuals with higher GC concentrations exhibit enhanced aggression or fear responses in various species [18–21]. Testosterone, a sex hormone belonging to the androgen family, is positively correlated with aggression [22–26]. Notably, cortisol and testosterone interactively regulate aggressive behaviors [27,28]. In addition, the cortisol concentrations in domestic cats are lower than those in wild cats [29]. Furthermore, a direct association has been established between cortisol and aggression in feral female cats [30]. A previous study has also shown that cortisol can increase testosterone secretion in vitro [31]. However, to date, no survey has explored the components of social behavior that may be related to GC concentrations in group-living cats. Oxytocin, a peptide hormone, is well-known for its role in influencing reproductive behaviors, such as mating and maternal care, and regulates diverse social behaviors related to 'tend-and-defend' within a group [32–34]. Therefore, if domestic cats co-habit as 'groupmates,' oxytocin would probably induce a 'tending' behavior among them.

Changes in food resource niches can modulate gut microbiome [35]. Incidentally, there exists a brain-gut-axis, in which gut microbes interact with the gut as well as the brain; one of the pathways via which gut microbes influence brain function is through endocrine activity, such as GCs, sex steroids, and neuropeptides [36]. The microbes can produce these hormones directly [37,38] or indirectly [39]. Notably, the microbiome has a great influence on the hypothalamic-pituitary-adrenal (HPA) axis [40,41] and the hypothalamic-pituitary-gonadal (HPG) axis [42]. Recent empirical studies have revealed that the gut microbiome can alter host sociality by modulating oxytocin secretion from the hypothalamus [43]. Therefore, changes in a host's niche can modulate gut microbiome and influence host sociality via such endocrine pathways.

Another notable finding in this context is the similarities in gut microbiome among groupmates that can arise due to microbial transmission through social interactions, or via shared environments in human and non-human primates [44–46]. A microbiome can generate biochemical signals that are used in the host's social communications in insects such as drosophila [47–49], or it can directly affect the host's nervous system in rodents [50–52], which, in turn, can affect host social behavior. Such observations suggest that the exchange of microbiomes and biochemical similarity can enhance group formation, resulting in close connections within groups.

The aim of the present study was to investigate the nature of interactions among cats within a group, and how such interactions are associated with various internal factors. Hence, we analyzed the relationships between the gut microbiome and concentrations of different hormones, such as cortisol, testosterone, and oxytocin, and the potential associations with social behavior of the group-living cats. Our four hypotheses were as follows: 1. individuals with high cortisol and testosterone concentrations will be less socially tolerant, and they will exhibit more aggression-related behaviors than affiliative behaviors; 2. individuals with high oxytocin concentrations will exhibit more affiliative behaviors than aggressive responses; 3. there is a relationship between the composition of gut microbiomes and hormone concentrations in an individual's body; 4. the gut microbiomes of individuals that have frequent contact among them are similar. Incidentally, to the best of our knowledge, this is the first attempt to reveal the role of oxytocin in conspecific social behaviors in a solitary mammal.

## Methods

### Subjects

Cats living in a shelter (Tanpoponosato, Kanagawa, Japan) participated in this experiment. There were 10 males and five females, with a mean age of 4.2 ± 2.3 years. All the cats were neutered, and they were divided into three groups of five cats each, randomly. All cats were housed in one room at Azabu University during the experiment. Further details regarding the participating cats are provided in Supplementary Information (S1 Table).

### Experimental setting

Each group of five cats was housed in a room (4 m × 7.5 m) for two weeks. More than five beds were set up in the room so that the cats could choose where to rest. There were five litter trays, two food bowls, and two water bowls. The cats were able to eat food and drink water at any time. Two cameras (HX-A1H, Panasonic, Japan) and two infrared lights were set up at the top corners of the room. The infrared lights were switched on during the night, enabling observations in the dark. Every alternate day, the behavior of the cats from 21:00 h at night to 7:00 h the next morning were analyzed. Their urine and feces samples were collected every morning (between 7:00 h and 10:00 h). The experiment was conducted one group at a time, and the three groups stayed in the same room. In addition, all the cats ate the same food. All cats were fed a complete and balanced commercially available kibble diet. The composition of the cat's gut microbiome was not affected by room environment or food. All protocols are carried out in accordance with relevant guidelines and regulations. All experimental procedures were approved by the Animal Ethics Committee of Azabu University (#180410–1).

### Behavioral analysis

Based on observations of the first group throughout the day, interaction among cats was quite low during the day, and they were more active at night. Therefore, behavioral analyses were henceforth conducted every alternate day from 21:00 h to 7:00 h the next morning for two weeks (70-h observation for each cat). We focused on each cat and recorded its behaviors and those of partners. The recorded behaviors and their definitions are listed in Supplementary Information (S2 Table). Active (subject cat initiating the behavior) and passive (subject cat receiving the behavior) behaviors were marked. BORIS v.7.0.8 (https://www.boris.unito.it/) was used for behavioral analysis. To check the inter-observer reproducibility, the observers analyzed the same video for 30% of the total video time. The kappa coefficient, which indicates reproducibility, was 0.86.

## Assay of urinary hormone concentrations

The urine samples of the cats were collected using a two-tiered litter box immediately after urination. The collected urine samples were centrifuged for 15 min (at 4°C and 3000 rpm) and immediately stored in a freezer at −80°C. Sixty-three urine samples, i.e., 4.2 ± 2.4 samples/individual (A minimum of one and a maximum of nine urine samples were collected per individual) were collected during the entire observation period, and all samples were assayed for cortisol concentrations. Oxytocin and creatinine concentrations were analyzed in 62 samples (4.1 ± 2.4 samples/individual); one sample was excluded due to its low quantity. Testosterone concentrations were assayed for 57 samples (3.8 ± 2.0 samples/individual). The assay protocols were based on previous studies [53–56]. The average of the urine hormone concentrations for each individual was used as the hormone baseline for each individual.

**Cortisol concentrations.** Cortisol concentrations were measured using enzyme-linked immunosorbent assay (ELISA). The undiluted urine samples were dispensed into the wells of the ELISA-plate. The primary antibody was anti-cortisol antibody (ab1949; Abcam Plc., UK) that had been diluted 200,000-fold, and the secondary antibody was mouse IgG-Fc fragment antibody (A90-131A, Bethyl Laboratories, USA) that had been diluted 500-fold. The horseradish peroxidase (HRP) used in this case was Cortisol-3-CMO-HRP (FKA403, Cosmo bio, Japan) that was diluted 1 million-fold. Standard and urine samples were dispensed in 15-μL volumes, and primary antibody, secondary antibody, and HRP in 100-μL volumes. The obtained values of the cortisol concentrations were adjusted for creatinine correction.

**Testosterone concentrations.** Testosterone concentrations were also measured using the ELISA technique. Based on the concentration, either undiluted urine samples were dispensed into the wells of the ELISA-plate, or the samples were diluted two-fold with a phosphate buffer containing 0.1% bovine serum albumin, before loading. The primary antibody was anti-testosterone 3 CMO antibody (ab35878, Abcam Plc., UK) that had been diluted 25,000-fold, and the secondary antibody was mouse IgG-Fc fragment antibody (A90-131A, Bethyl Laboratories, USA) that had been diluted 500-fold. The HRP used in this process was Testosterone-3-CMO-HRP (FKA101, Cosmo bio Co., Ltd., Japan) that was diluted 1 million-fold. Standard and urine samples were dispensed in 25-μL volumes, and primary antibody, secondary antibody, and HRP in 100-μL volumes. The obtained values of the testosterone concentrations were adjusted for creatinine correction.

**Oxytocin concentrations.** The commercially available oxytocin ELISA kit (ADI-901-153A-0001, Enzo Life Sciences, Inc., USA) was used for the assay. The urine samples were diluted 50-fold with the assay buffer in the kit and dispensed into the wells of the ELISA-plates. Standard and urine samples were dispensed in 15-μL volumes, and primary antibody and HRP in 50-μL volumes. The obtained values of the oxytocin concentrations were adjusted for creatinine correction.

**Creatinine concentrations.** The creatinine standard and the urine samples were diluted 100-fold with distilled water and dispensed into a 96-well microplate (AS ONE Co., Ltd., Osaka) at 100 μL each, followed by 50 μL of 1 M NaOH and 50 μL of 1 g/dL trinitrophenol. The plate was left at room temperature (22–26°C) for 20 min, and the absorbance was measured at a wavelength of 490 nm using a microplate reader (MODEL 680XR, Bio-Rad Laboratories, Inc., USA). The samples were used undiluted.

## Analysis of gut microbes

Fecal samples were collected from different individuals only when the cats that excreted them were observed. The fecal samples were collected from only eight individuals. For the first group of cats, the feces were collected within 3 h of defecation, stored in a refrigerator at 4°C,

processed into a glycerol stock within 24 h, and stored in a −80˚C freezer. For the second and third groups, approximately 1.0 g of feces was collected within 15 min of defecation using a sterile cotton swab; thereafter, it was placed in a 15 mL tube containing a reagent mixture of 1.0 mL phosphate-buffered saline and 2.0 mL glycerol and dissolved using a bamboo skewer. Although different collection methods were used, the time of the sampling did not change the main composition of the microbiome as long as we collected the part of the feces that is not exposed to the air near the center (Sinha et al. 2016). The use of the swab did not change the composition of the microbiome (Sinha et al. 2016). It was then stored in a −80˚C freezer. Subsequently, we transferred the intact feces into a tube containing 2.5 mm-diameter cell-grinding beads and analyzed gut microbiomes at the Bioengineering Lab. Co., Ltd.

DNA was extracted from the crushed sample using the MPure Bacterial DNA Extraction Kit (MP Bio Japan, Japan). Then, the concentration of the extracted DNA solution was measured using Synergy H1 (BioTek) and QuantiFluor dsDNA System, and the library was prepared. The library was prepared using the two-step tailed PCR method. The V3–V4 region of the 16S ribosomal RNA (rRNA) gene was amplified using PCR. In the 1st PCR, a solution was prepared by adding 1.0 μl of 10XEx Buffer, 0.8 μl of deoxynucleotide Triphosphates (dNTPs, 2.5mM each), 0.5 μl of Forward primer (10 μM), 0.5 μl of Reverse primer (10 uM), 2.0 μl of Template DNA (0.5 ng/μl), 0.1 μl of ExTaqHS (TaKaRa) (5 U/μl), and 5.1 μl of deionized distilled water (DDW). The 16S Amplicon PCR Forward Primer (5′–ACACTCTTTCCCTACAC GACGCTCTTCCGATCT–NNNNN– CCTACGGGNGGCWGCAG–3′) and 16S Amplicon PCR Reverse Primer (5′–GTGACTGGAGTTCAGACGTGTGCTCTTCCGATCT–NNNNN–GACTAC HVGGGTATCTAATCC–3′) were used [57]. PCR amplification was performed by pre-denaturation at 94˚C for 2 min, followed by 30 cycles of 94˚C for 30 s, 55˚C for 30 s, 72˚C for 30 s, and final extension at 72˚C for 5 min. The PCR product was washed using AMPure XP beads (BECKMAN COULTER). In the 2nd PCR reaction, a solution was prepared by adding 1.0 μl of 10XEx Buffer, 0.8 ul of dNTPs (2.5mM each), 0.5 μl of Forward primer (10μM), 0.5 μl of Reverse primer (10μM), 2.0 μl of PCR product (max 5 ng/ul), 0.1 μl of ExTaqHS (TaKaRa) (5U/μl), and 5.1 μl of DDW. The 16S Amplicon PCR Forward Primer (5′–AATGATACGGC GACCACCGAGATCTACAC–Index2–ACACTCTTTCCCTACACGACGC–3′) and 16S Amplicon PCR Reverse Primer (5′–CAAGCAGAAGACGGCATACGAGAT–Index1–GTGACTGGA GTTCAGACGTGTGT–3′) were used. PCR amplification was performed by pre-denaturation at 94˚C for 2 min, followed by 10 cycles of 94˚C for 30 s, 60˚C for 30 s, 72˚C for 30 s, and final extension at 72˚C for 5 min. The PCR product was washed using AMPure XP beads (BECKMAN COULTER). The quality of libraries was evaluated using a Fragment Analyzer and a dsDNA 915 Reagent Kit (Advanced Analytical Technologies, Inc., USA) [58]. The purified products were sequenced using the Illumina MiSeq System (Illumina Inc., San Diego, CA, USA). We obtained 428,762 total reads (53,595 ± 4589).

For data analysis, we extracted only those sequences whose start reading was an exact match to the primer used, using fastq_barcode_spliltter in the Fastx toolkit [59]. The primer sequences of the extracted sequences were removed. We then used sickle tools to remove sequences with a quality value < 20, and discarded sequences < 150 bases in length and their paired sequences. The paired-end merge script FLASH [60] was used to merge the sequences that passed the quality filtering. The merging conditions were 420 bases for the merged fragment length, 280 bases for the read fragment length, and 10 bases for the minimum overlap length. The sequences that passed all the filtering were checked for chimeras using the uchime algorithm of usearch [61]. The reads that were not determined to be chimeras were used in subsequent analyses. In addition, if there were non-bacterial reads, they were removed. The creation of OTUs and phylogenetic estimation were performed using QIIME workflow scripts with no reference and all parameters set to default conditions. The number of reads used in

the QIIME analysis was 224,878 (28,110 ± 1764). The genetic data were registered in DNA Data Bank of Japan (https://www.ddbj.nig.ac.jp/index.html: 191 accession number PRJDB12856).

## Statistical analysis

Spearman's rank correlation coefficient was used to examine the correlation between the hormonal concentrations and the behaviors of the cats, and different hormones. To investigate the similarity in gut microbes among the cats, a hierarchical cluster analysis was performed based on the percentage of similar microbes in the cats at the genus level. In addition, the UniFrac distance was calculated by constructing a phylogenetic tree using representative sequences from each OTU and UniFrac Principal Coordinate Analysis (PCoA) calculated. To assess the relationship between gut microbiome similarity and contact frequency among the cats, the correlation between weighted UniFrac distance and each behavior was examined using the Spearman's rank correlation coefficient. The correlations between the components of each axis by PCoA of gut microbe OTUs and hormones were analyzed. In addition, the Mann-Whitney test was used to examine the sex-based differences in hormone concentrations, and Spearman's rank correlation coefficient was used to examine the associations between hormone concentrations and age. JMP v14.2.0 (SAS Institute, Cary, NC, US) was used for all analyses.

# Results

## Correlation between hormone concentrations and behaviors of cats

**Cortisol.** We observed negative correlations between cortisol concentration and contact among cats, as well as the food sharing behavior (S3 Table). Significant negative correlations (p < 0.05) were specifically observed between cortisol concentration and active following (rs = −0.556, p = 0.032), playing (active: rs = −0.580, p = 0.023; passive: rs = −0.679, p = 0.005), sharing of food (rs = −0.681, p = 0.005), and passive sniffing (rs = −0.539, p = 0.038).

**Testosterone.** The results of correlation between testosterone concentration and cat behaviors are presented in S4 Table. Significant positive correlations (p < 0.05) were observed between testosterone concentration and active escape behavior (rs = 0.560, p = 0.030).

**Oxytocin.** We observed negative correlations between oxytocin concentration and contact among cats, as well as food sharing behavior, similar to in the results for cortisol (S5 Table). Significant negative correlations (p < 0.05) were observed between oxytocin concentration and active allo-grooming (rs = −0.678, p = 0.006), passive following (rs = −0.660, p = 0.007), passive playing (rs = −0.597, p = 0.019), sharing of food (rs = −0.597, p = 0.034), and sniffing (active: rs = −0.596, p = 0.019; passive: rs = −0.761, p = 0.001).

Furthermore, we analyzed correlations among concentrations of the different hormones (Fig 1). There was a significant positive correlation (p-value < 0.05) between concentrations of cortisol and testosterone (Fig 1A; rs = 0.636, p = 0.011). Incidentally, there was no effect of sex (S1 Fig; cortisol: Z = 0.412, p = 0.680; testosterone: Z = −0.177, p = 0.860; oxytocin: Z = 0.503, p = 0.596) or age (S2 Fig; cortisol: rs = 0.265, p = 0.361; testosterone: rs = −0.147, p = 0.615; oxytocin: rs = 0.087, p = 0.766) on any of the hormones.

## Gut microbes

**Similarity of the proportion of gut microbes among the cats.** First, to investigate the compositions of the gut microbiomes of individuals, the hierarchical clusters of the proportion of gut microbes observed at the genus level of the cats are shown in Supplemental Information

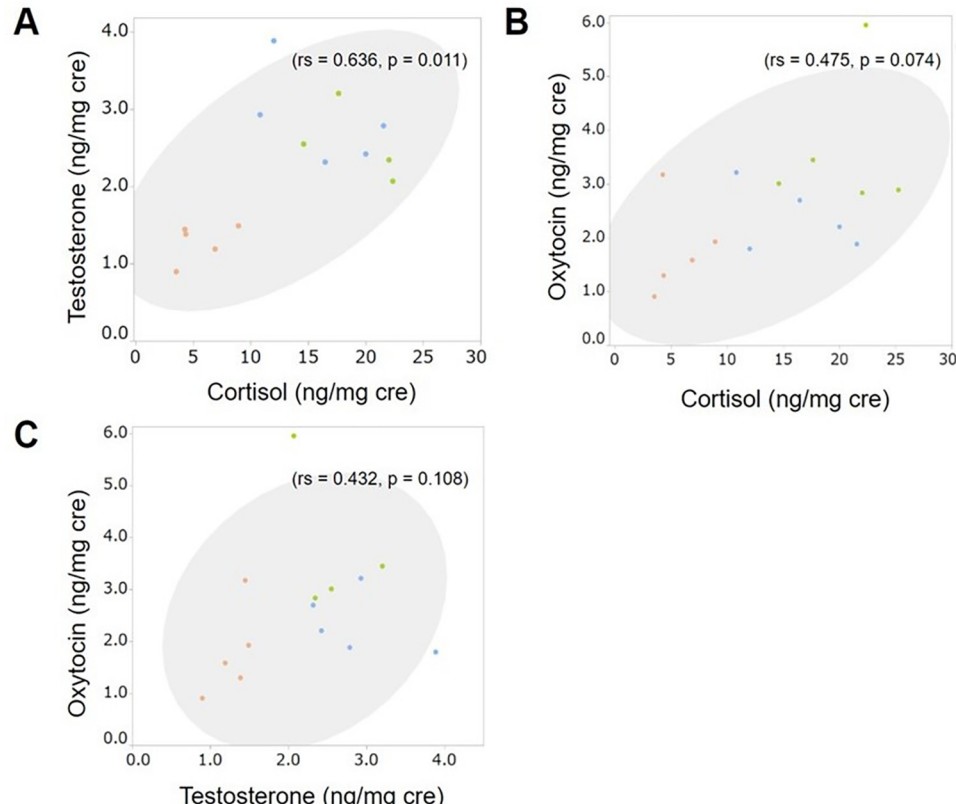

**Fig 1. Correlations among different hormones (cortisol, testosterone, and oxytocin).** Fig 1A shows the correlation between cortisol and testosterone, 1B shows the correlation between cortisol and oxytocin, and 1C shows the correlation between testosterone and oxytocin. Each group is color coded. Each point represents the average hormone concentration for each individual. The gray circle is a 95% confidence ellipse.

(S3 Fig). Next, to investigate whether the compositions of the microbiomes are affected by contact with other individuals, we analyzed the correlations among UniFrac distances, which indicates similarity of gut microbiomes between individuals, and the interactions among the cats. Negative correlations were observed with respect to some of the behaviors, namely sharing a bed, entering bed, grooming, and sniffing (S6 Table). Hence, the greater the similarity in gut microbiome, the greater the sharing bed (Fig 2A, rs = −0.679, p < 0.001), entering bed (Fig 2B, rs = −0.395, p = 0.037), and sniffing (Fig 2C, rs = −0.446, p = 0.018).

**Correlations of gut microbiome with hormone concentrations as well as behaviors of the cats.** The correlations between each component of the axis based on PCoA of the microbes and hormone levels in the cats (S7 Table) as well as those between the PCoA and cat behavior (S8 Table) were analyzed. PCoA2 was correlated negatively with cortisol concentration (rs = −0.714, p = 0.047) and positively with rubbing (total: rs = 0.819, p = 0.013), following (total: rs = 0.747, p = 0.033), sharing of food (rs = 0.733, p = 0.039), and sniffing (rs = 0.738, p = 0.037) behaviors. Additionally, PCoA3 was correlated positively with entering bed (active: rs = 0.762, p = 0.028; passive: rs = 0.781, p = 0.022; total: rs = 0.762, p = 0.028), and was correlated negatively escaping behavior (rs = −0.903, p = 0.002). PCoA5 was correlated positively with cortisol concentration (rs = 0.762, p = 0.028), and negatively with following behavior (active: −0.866, p = 0.005). Moreover, PCoA8 was correlated with attacking behavior (rs = −0.714, p = 0.047).

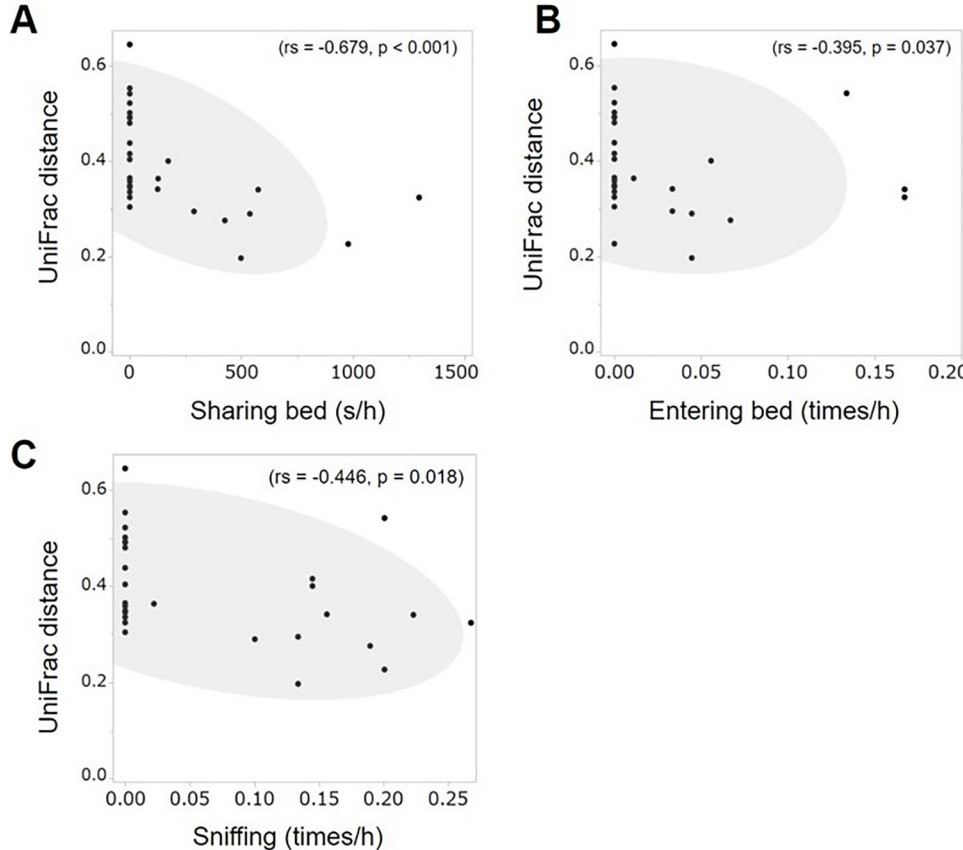

**Fig 2. Correlations among cat behaviors and UniFrac distances of their gut microbiomes.** Fig 2A shows the correlation between 'sharing bed' (s/h) and UniFrac distance, 2B shows the correlation between 'entering bed' (times/h) and UniFrac distance, and 2C shows the correlation between 'sniffing' (times/h) and UniFrac distance. The points indicate data in each pair, e.g., Cat 10 and Cat 11, Cat 10 and Cat 12. The gray circle is a 95% confidence ellipse.

## Discussion

At the beginning of the study, it was hypothesized that cats with high concentrations of cortisol and testosterone would be less tolerant in their social interactions. The results of our study are consistent with our hypothesis. We observed that cats with high cortisol concentrations as well as those with high testosterone concentrations had less contact with other individuals; moreover, cats with high testosterone concentrations had a greater tendency to escape than cats with low testosterone concentrations. However, the correlation between oxytocin concentration and contact behaviors among cats was inconsistent with our hypothesis that high oxytocin concentrations would make them socially affiliative. The result is the first to reveal the correlation between cat–to–cat interaction and oxytocin concentration. Furthermore, cats with similar gut microbiomes exhibited high degrees of interaction with other cats during the study period, and correlations were also observed between gut microbiomes of the cats and their behavioral patterns, as well as cortisol concentrations.

### Hormones and social behaviors

Previous studies have reported positive correlations between oxytocin concentration and affiliative contact with partners, such as allo-grooming, in several group-living species

[32,33,62,63]. On the contrary, in this study, we observed a negative correlation between oxytocin concentration and affiliative contact among the cats living in the same space. Oxytocin functions in an affiliative manner, with respect to in-group individuals, but in an exclusionary manner, for out-group members [34,62], which implies that individuals with high oxytocin concentrations rarely exhibit affiliative behaviors with out-group members. Based on the results, even if cats spend time together and share the same space, they might not be able to form tightly connected groups since each cat might be consider the other cats out-group individuals. The experimental period of two weeks may be too short to enable the formation of a tight relationship group and observing them for a longer period could result in a change in their relationships. Although their relationships were possibly established since the cats who participated in the experiment had been living in the same room at the shelter before the experiment, investigations over a longer duration could offer greater insights on group-formation in cats.

Notably, the function of oxytocin varies between species. A previous study showed that eye contact in response to oxytocin is different between chimpanzees and bonobos—while in bonobos oxytocin increases eye contact, in chimpanzees oxytocin appears to have the opposite effect, and reduce eye contact [64]. The characteristics of species and the ecological environment surrounding them may cause different functions of oxytocin. It is possible that the characteristics of the cats in the experiment, such as sex and age, as well as the ecological environment in which they have been living, influence oxytocin function.

In the present study, we observed that cats who made contact with the other group members without any fear had a lower cortisol concentration. The results are consistent with those of studies on silver fox domestication, in which individuals were bred to show no fear or anxiety reactions towards humans [21,65]. Domesticated individuals with an advanced selection for being human-friendly exhibited higher tolerance toward human approach, and their GC secretion was inhibited. Therefore, the selection for fearlessness led to the inhibited development of the HPA axis, which is consistent with our results. Hence, the inhibition of the GC secretion system most probably contributed to the cohabitation as well as the high tolerance toward the group members in cats. In future, further studies may be performed to elucidate the biological mechanisms underlying such correlation.

Several previous studies have demonstrated that testosterone is associated with aggression in some species [22–25]. Although we did not find any association between testosterone and aggressive behavior in the cats, the individuals with high testosterone concentrations showed a high tendency to escape. The results also suggest that in spayed and castrated cats, testosterone is involved in developing tolerance towards other individuals, even though it does not lead to any aggressive behavior. Moreover, due to castration, the testosterone concentrations declined rapidly in the male cats [66]; hence, there was no difference between testosterone concentrations in the two sexes in the present study. Despite castration, testosterone secretions from the adrenal cortex may influence the temperaments of individual cats from this result.

In the present study, we investigated the relationship between basal hormone levels and cat behavior. Previously, studies have reported that several hormones are responsive in nature, especially cortisol [67]. Moreover, the psychological factors of an individual, such as implicit power motive and coping style [68–70], as well as the behavioral ecology [71] of the species can influence the hormonal variations with respect to a certain behavior. For instance, in the case of bonobos and chimpanzees, the behavioral ecology of the species leads to differences in endocrine changes with respect to competition [14]. Therefore, in future, the investigation of responsive hormones may offer a more comprehensive understanding of the influence of the behavioral ecology of cats on the relationship between behavior and basal hormone concentrations.

## Gut microbiome

In the present study, the more interactions there were between individuals, the more similar their gut microbiome was. When animals are housed together and the environment is shared, there is increased similarity in their gut microbes owing to the increased chances of direct or indirect contact [72]. Previously, a study on baboons demonstrated that microbes may be transmitted through contact [44]. Incidentally, allo-grooming, which involves the exchange of biochemicals from external secretions, has a stronger influence on transmission of gut microbes than any other physical contact behavior. However, we did not observe any relationship between allo-grooming and similarity in the gut microbes in the cats. This could be due to the small sample size and limited time frame of the study. In addition, this study examined the gut microbiome of cats that stayed in the same room and ate the same food, but it did not completely eliminate the effects of environment, because the experiments were conducted at different times. Therefore, it needs to be re-examined using the time series gut microbiomes of a larger sample size living in the same room at the same period. A key question to be addressed is whether the sharing of microbiomes through physical contact produces a fitness benefit or cost to the host. In this respect, there is a positive aspect in which the microbiome can recolonize in individuals that have lost beneficial microbes due to disease or use of antibiotics, through microbiome transmission from other members. On the contrary, there is a negative aspect in which highly interactive individuals have high chances of being exposed to pathogenic microbes.

Gut microbes reportedly affect brain function, thereby causing phenotypic changes in individuals, such as behavioral alterations. Gut microbes are involved in the development of the HPA and HPG axes [40,42] and generation of oxytocin-expressing neurons [43]. One of the pathways via which the gut microbes influence brain functions is through endocrine activity, such as secretion of GCs, testosterone, and oxytocin. Since we observed a correlation between gut microbes and cortisol secretion, as well as between gut microbes and behavioral responses of the cats, it is highly likely that the gut microbes influence hormone secretions and behavior mechanisms of individuals. However, in the current study, we did not analyze the relationship between the microbial species and behavioral responses or hormone concentrations in detail. In future, an extensive research with a large sample size and a detailed analysis, together with empirical experiments using germ-free mice, may reveal which microbes specifically influence behavior and hormone concentrations. Most mammalian gut microbiome studies have focused on fecal microbiota, but it is unclear how well fecal samples reflect the microbiota of the intestinal region, and these should also be examined.

## Limitation and future perspective

A follow-up study observing the cats over a period of several months might provide more comprehensive information. In addition, our study was limited to correlations between hormones and behaviors, so that causality is not known. Since the subject cats were of different or unknown ages and backgrounds, it may have been impossible for them to live as 'group mates'. Hence, in future studies, it is necessary to examine the factors that can affect the ability of the cats to consider each other as 'in-group' members and form affiliative relationships among them; for instance, the experience of spending time together from their juvenile period can facilitate tight group formation among the cats [73]. Moreover, the sex composition of the members of groups was not consistent. Experiments with male-only and female-only groups could facilitate the elucidation of the adaptive significance of gender in cats living in groups.

Hormones regulate social behavior by binding to their specific receptors in the brain. Hormones can affect social behavior in different ways, depending on the number of receptors as well as the region of the brain where the receptors are present. For example, the expression patterns of

oxytocin receptors differ greatly between prairie voles (*Microtus ochrogaster*) and montane voles (*Microtus montanus*), which has been suggested to be one the factors influencing their monogamous/polygamous nature [74]. Therefore, future studies should focus on the expression patterns of hormone receptors in domesticated cats to uncover the sociality of the cats living in groups.

In summary, we observed that cortisol, testosterone, and oxytocin concentrations are correlated with the social behavior of the cats, which are considered solitary animals, and the gut microbial composition is related to social behavior as well as endocrine concentrations. Changes in the endocrine system may lead to temperament changes, which, in turn, enable the cats to share space with other cats. Furthermore, this study sheds new light on the role of oxytocin in solitary animals, and further research is required to understand the mechanisms underlying such a relationship. Since behavior is greatly influenced by external environmental factors, such as temperature and food resources, examining changes in social relationships among cats that occur as a result of variations in the external environment could provide further insights into the factors that influence sociality among the individuals.

## Supporting information

**S1 Fig. Sex differences in cortisol, testosterone, and oxytocin concentrations.** S1A Fig indicates cortisol, S1B indicates testosterone, and S1C indicates oxytocin. Each point shows average hormone concentration for each individual.
(TIF)

**S2 Fig. Correlation between age and cortisol, testosterone, and oxytocin concentrations.** S2A Fig indicates cortisol, S2B indicates testosterone, and S2C indicates oxytocin. Each point shows average hormone concentration for each individual. The gray circle is a 95% confidence ellipse.
(TIF)

**S3 Fig. Similarity in the proportion of gut microbes.** The hierarchical clusters are shown based on the percentages of gut microbes at the genus level. The colors of each cat are color-coded for each group (Orange: Group 1, Blue: Group 2, Green: Group 3).
(TIF)

**S1 Table. Profile of cats.** This table shows each group of cats, their sex, and age. The colors are divided by groups, with the first group shown in orange, the second group in blue, and the third group in green.
(CSV)

**S2 Table. Analyzed behaviors and their definitions.**
(CSV)

**S3 Table. Correlation between cortisol concentrations and behaviors of cats.** The results of Spearman's rank correlation coefficient between cortisol concentrations and cat behaviors are listed. The letters in brackets following each behavior are: a for active, p for passive, and t for the sum of passive and active behavior. Those with p-values $< 0.05$ have been marked in bold. The data are color-coded according to the value of the correlation coefficient (Blue: $-0.8$ to $-0.6$, Light blue: $-0.6$ to $-0.4$, Light orange: 0.4 to 0.6, Orange: 0.6 to 0.8). The rs are correlation coefficients.
(CSV)

**S4 Table. Correlation between testosterone concentrations and behaviors of cats.** The results of Spearman's rank correlation coefficient between cortisol concentrations and cat

behaviors are listed. The letters in brackets following each behavior are: a for active, p for passive, and t for the sum of passive and active behavior. Those with p-values < 0.05 have been marked in bold. The data are color-coded according to the value of the correlation coefficient (Blue: −0.8 to −0.6, Light blue: −0.6 to −0.4, Light orange: 0.4 to 0.6, Orange: 0.6 to 0.8). The rs are correlation coefficients.
(CSV)

**S5 Table. Correlation between oxytocin concentrations and behaviors of cats.** The results of Spearman's rank correlation coefficient between oxytocin concentrations and the cat behaviors are listed. The letters in brackets following each behavior are: a for active, p for passive, and t for the sum of passive and active behavior. Those with p-values < 0.05 are shown in bold. The data are color-coded according to the value of the correlation coefficient (Blue: 521 −0.8 to −0.6, Light blue: −0.6 to −0.4, Light orange: 0.4 to 0.6, Orange: 0.6 to 0.8). The rs are correlation coefficients.
(CSV)

**S6 Table. Correlations of UniFrac distances of gut microbiome with interactions among the individuals.** Those with p-values < 0.05 are shown in bold. For sharing food and playing, statistical processing was not possible due to the small number of individuals that exhibited the behaviors. The rs are correlation coefficients.
(CSV)

**S7 Table. Correlations of the components of each component of the axis based on principal coordinate analysis (PCoA) axis with cortisol, oxytocin, and testosterone concentrations.** Those with p-values < 0.05 are shown in bold. The rs are correlation coefficients.
(CSV)

**S8 Table. Correlations of the components of each component of the axis based on principal coordinate analysis (PCoA) axis with the behaviors of the cats.** The letters in brackets following each behavior are: a for active, p for passive, and t for the sum of passive and active behavior. Those with p-values < 0.05 are shown in bold. The rs are correlation coefficients.
(CSV)

**S9 Table. Dataset_Behaviors and hormones.**
(CSV)

**S10 Table. Dataset_List of operational taxonomic units (OTUs), which are classification units in clustering based on 16SrRNA gene sequences.**
(CSV)

**S11 Table. Dataset_Weighted unifrac distance and behaviors.**
(CSV)

**S12 Table. Dataset_Weighted unifrac pcoa and hormones and behaviors.**
(CSV)

## Acknowledgments

We would like to thank the cat shelter "Tanpoponosato" for their cooperation in the experiment.

## Author Contributions

**Conceptualization:** Hikari Koyasu, Hironobu Takahashi, Moeka Yoneda, Syunpei Naba, Natsumi Sakawa, Ikuto Sasao, Miho Nagasawa, Takefumi Kikusui.

**Data curation:** Hikari Koyasu.

**Formal analysis:** Hikari Koyasu.

**Funding acquisition:** Hikari Koyasu, Miho Nagasawa, Takefumi Kikusui.

**Investigation:** Hikari Koyasu, Hironobu Takahashi, Moeka Yoneda, Syunpei Naba, Natsumi Sakawa, Ikuto Sasao.

**Methodology:** Hikari Koyasu, Moeka Yoneda, Syunpei Naba, Natsumi Sakawa, Ikuto Sasao, Miho Nagasawa.

**Project administration:** Miho Nagasawa, Takefumi Kikusui.

**Supervision:** Miho Nagasawa, Takefumi Kikusui.

**Visualization:** Hikari Koyasu.

**Writing – original draft:** Hikari Koyasu, Miho Nagasawa, Takefumi Kikusui.

**Writing – review & editing:** Hikari Koyasu, Miho Nagasawa, Takefumi Kikusui.

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
