## [Decision Letter · Decision Letter 0]

26 Dec 2021

PONE-D-21-34568Correlation between behavior and hormone or gut microbiome implies that domestic cats (Felis silvestris catus) living in a group are not like ‘groupmates’PLOS ONE

Dear Dr. Koyasu,

Thank you for submitting your manuscript to PLOS ONE. After careful consideration, we feel that it has merit but does not fully meet PLOS ONE’s publication criteria as it currently stands. Therefore, we invite you to submit a revised version of the manuscript that addresses the points raised during the review process.

We look forward to receiving your revised manuscript.

Kind regards,

Chun Wie Chong

Academic Editor

PLOS ONE

Journal Requirements:

“This work was supported by the Japan Society for the Promotion of Science and Grants-in-Aid for Scientific Research from the Ministry of Education, Culture, Sports, Science, and Technology of Japan (Grant numbers: #20J14760 to H.K., 23 #18H02489 and #19K22823 to M.N., and #19H00972 to T.K.).”

“This work was supported by the Japan Society for the Promotion of Science (https://www.jsps.go.jp/) and Grants-in-Aid for Scientific Research from the Ministry of Education, Culture, Sports, Science, and Technology of Japan (https://www.mext.go.jp/) The grant numbers are #20J14760 to H.K., #18H02489 and #19K22823 to M.N., and #19H00972 to T.K.

Reviewers' comments:

Reviewer's Responses to Questions

**Comments to the Author**

1. Is the manuscript technically sound, and do the data support the conclusions?

Reviewer #1: Partly

Reviewer #2: Yes

Reviewer #3: Partly

2. Has the statistical analysis been performed appropriately and rigorously? 

Reviewer #1: N/A

Reviewer #2: Yes

Reviewer #3: Yes

3. Have the authors made all data underlying the findings in their manuscript fully available?

Reviewer #1: No

Reviewer #2: Yes

Reviewer #3: No

4. Is the manuscript presented in an intelligible fashion and written in standard English?

Reviewer #1: Yes

Reviewer #2: Yes

Reviewer #3: Yes

5. Review Comments to the Author

Reviewer #1: The manuscript addresses an interesting relationship between behaviour, hormones and gut microbiota of group-living cats. However, the methodology and statistics used in this study is not well-explained and lack sufficient detail to evaluate the relevance of the findings.

First, the analysis of gut microbiota section must have a detailed information about the laboratory and bioinformatic processing. Even if the processing was done by a company all the information about the processing steps need to be addressed. There is no information about extraction method, PCR settings, library building, primers, sequencing strategy, if biological or technical replicates were used, how many blanks and what kind were used… Moreover, the faecal samples were collected using two different method? Was this controlled in the analysis?

First of all, the gut microbiota analysis section should have detailed information about the laboratory and bioinformatics processing. Even if the processing was done by a company, all information about the processing steps must be addressed in the methodology. There is no information on the extraction method, PCR setup, library construction, primers, sequencing strategy, if biological or technical replicas were used, how many blanks and what type were used... Moreover, faecal samples were collected using two different methods. Was this controlled in the analysis?

Second, it is not sufficient to write that the “data were processed using QIMME”, the author should include detailed information about how the sequences were analysed (e.g. demultiplexing, primer removal, filtering, denoising, contaminant removal…). In addition, the authors should make the bioinformatic pipeline available.

Third, it is not clear to me how the faecal samples were collected. In the experimental setting section, the authors say that 8 faecal samples were collected, but from which individuals? Was it possible to identify who owned each stool sample? Otherwise, how was it possible to correlate a specific microbial community with a specific behaviour? And finally, have you considered in the analysis that some faecal samples belong to cats that share a room (5 rooms and 8 faecal samples)? Therefore, some samples are not independent. How was this controlled in the statistical analysis?

Fourth, since the relevant correlations (and p-values) are already included in the text and the tables do not provide any additional information, all the tables should be moved to supplementary material.

Lastly, the discussion part should be re-organized to clarify the findings of this study and its consequences.

Reviewer #2: This study provides insight between behavior of shelter cats correlated to hormonal and gut microbiome profiling. The conclusions obtained were valuable as the findings shed light on the hormonal status of originally solitary or group-living animals. It is hopeful that the conclusions drawn from this study can be an example to help behaviorists incorporate more measurable parameters which will reduce subjectivity when relying only on visual observations. Despite the gender-neutral assumption, it might be worthwhile to have gender-specific groupings to further understand its impact on behavior for future studies, in addition to increased sample size and longer duration. The methods used were sound and results obtained provided clarity on the hypotheses.

Reviewer #3: 1. Interesting. The authors addressed notable correlation between hormones and behaviours. While mounting studies had established the correlation between gut microbiome and behaviours, the present study lacking clarity in term of study design for determining the gut microbiota constituents. Microbiota composition varies inter-individually, and effected by various environmental factors. The recent study only sampled the cats stool at one point of time. It is suggested to sample the stool similarly like urine, and sequence the stool concurrently i.e. time series analysis for the gut microbiome.

2. The grouping strategy was vaguely described.

3. While the hormonal analyses were conducted for 50 plus samples, the microbiome analysis was conducted only on 8 cats subjects, with Group 1 only represented by 1 subjects.

4. Sample preparation (microbial genomics library preparation) was vaguely described.

5. Microbial genomics analysis was vaguely described.

6. The raw sequencing data were not publicly available. Data availability. We recommend that all data and related meta-data underlying the findings reported in a submitted manuscript will be deposited in an appropriate public repository (GenBank, European Nucleotide Archives, National Center for Biotechnology and Information, National Bioscience Data Center, DNA data bank of Japan).

7. Is there any specific reason to opt for UniFrac distance?

8. The manuscript was written in standard English, with minor grammatical errors.

9. Data presentation using PCoA were OK, but making individual comparison for microbiome analysis was vary confusing. I would suggest for the authors to conduct group comparison for the microbiome analysis. In fact, inter group correlation between behaviours also can be conducted.

6. PLOS authors have the option to publish the peer review history of their article (what does this mean?). If published, this will include your full peer review and any attached files.

Reviewer #1: No

Reviewer #2: No

Reviewer #3: No

---

## [Author Response · Author response to Decision Letter 0]

23 Jan 2022

24.01.2022

Chun Wie Chong

Academic Editor

PLOS ONE

Dear Dr. Chun Wie Chong

Thank you for inviting us to submit a revised draft of our manuscript (PONE-D-21-34568) titled “Correlation between behavior and hormone or gut microbiome implies that domestic cats (Felis silvestris catus) living in a group are not like ‘groupmates’”. We appreciate the time and effort you and each of the reviewers have dedicated to providing insightful feedback on how to strengthen our paper. Thus, it is with great pleasure that we resubmit our article for further consideration. We have incorporated changes that reflect the detailed suggestions you have graciously provided. We hope that our edits and the responses we provide below satisfactorily address all issues and concerns you and the reviewers have noted. Please find our point-by-point responses to the reviewers' comments below.

I am looking forward to hearing from you.

Sincerely,

Hikari Koyasu

Human-Animal Interaction and Reciprocity Laboratory,

Azabu University 

 

Reviewer #1: 

The manuscript addresses an interesting relationship between behaviour, hormones and gut microbiota of group-living cats. However, the methodology and statistics used in this study is not well-explained and lack sufficient detail to evaluate the relevance of the findings.

Reply: We thank the reviewer for all the detailed comments and suggestions. We found them very useful as we approached our revision.

First, the analysis of gut microbiota section must have a detailed information about the laboratory and bioinformatic processing. Even if the processing was done by a company all the information about the processing steps need to be addressed. There is no information about extraction method, PCR settings, library building, primers, sequencing strategy, if biological or technical replicates were used, how many blanks and what kind were used… Moreover, the faecal samples were collected using two different method? Was this controlled in the analysis?

Reply: We have added detailed information about the bioinformatic processing of gut microbiomes (lines 146-175). 

In addition, as mentioned by the reviewer, the fecal samples were collected using two methods, and it was best to collect them in one consistent way. The sampling method for the first group was for 16S rRNA analysis, in addition to the storage of bio-active microbiomes (glycerol stock). However, the application of this method would have involved high efforts and exceeded our storage space, so that we changed the collection method. The time of the sampling did not change the composition of the microbiome as long as we collected the part of the feces that was not exposed to the air near the center (Sinha et al. 2016). Moreover, the use of the swab did not change the composition of the microbiome (Sinha et al. 2016). Therefore, targeting a small number of bacteria in the fecal microbiome that are sensitive to oxygen may not eliminate the effect of differences in sampling methods, but by comparison of the overall structure, we determined that there were small effects on the analysis. 

Second, it is not sufficient to write that the “data were processed using QIMME”, the author should include detailed information about how the sequences were analysed (e.g. demultiplexing, primer removal, filtering, denoising, contaminant removal…). In addition, the authors should make the bioinformatic pipeline available.

Reply: We have added bioinformatic information and information about the analysis of the gut microbiome（lines 176-191).

Third, it is not clear to me how the faecal samples were collected. In the experimental setting section, the authors say that 8 faecal samples were collected, but from which individuals? Was it possible to identify who owned each stool sample? Otherwise, how was it possible to correlate a specific microbial community with a specific behaviour? And finally, have you considered in the analysis that some faecal samples belong to cats that share a room (5 rooms and 8 faecal samples)? Therefore, some samples are not independent. How was this controlled in the statistical analysis?

Reply: We collected the fecal samples from different individuals only when we observed which cats excreted the feces. Moreover, the experiment was conducted for one group at a time, and the three groups stayed in the same room. In addition, all cats were eating the same food. We have reduced the influence of the similarity of the gut microbiome through the room environment and food as much as possible. We have rewritten the method description on the fecal collection and experimental setting (lines 146, 96-98), and individuals for which fecal samples were collected are shown in Table S1 to address the reviewer’s comment. We hope that the revised section is now clarified.

However, it was not possible to completely eliminate the effects of the environment because each group was in the specific environment at different times. Since we were only able to collect samples from one cat in the first group and two cats in the second group, it was difficult to take into account that there were different groups in this study. Therefore, we have added to the discussion that it is necessary to examine the gut microbiome of more cats living in the same room at the same time (lines 325-329).

Fourth, since the relevant correlations (and p-values) are already included in the text and the tables do not provide any additional information, all the tables should be moved to supplementary material.

Reply: We thank the reviewer for the suggestion. All tables have been moved to the supplementary material.

Lastly, the discussion part should be re-organized to clarify the findings of this study and its consequences.

Reply: We have rewritten the discussion to clarify the findings and their consequences.

Reviewer #2: 

This study provides insight between behavior of shelter cats correlated to hormonal and gut microbiome profiling. The conclusions obtained were valuable as the findings shed light on the hormonal status of originally solitary or group-living animals. It is hopeful that the conclusions drawn from this study can be an example to help behaviorists incorporate more measurable parameters which will reduce subjectivity when relying only on visual observations. Despite the gender-neutral assumption, it might be worthwhile to have gender-specific groupings to further understand its impact on behavior for future studies, in addition to increased sample size and longer duration. The methods used were sound and results obtained provided clarity on the hypotheses.

Reply: We thank the reviewer for the useful comments. We agree on the importance of conducting further research with gender-specific groupings. The differences in frequency of behaviors that occur within males and females, and between males and females should be investigated by addition of further cats. We have added this information to the discussion (lines 351-352).

Reviewer #3: 

1. Interesting. The authors addressed notable correlation between hormones and behaviours. While mounting studies had established the correlation between gut microbiome and behaviours, the present study lacking clarity in term of study design for determining the gut microbiota constituents. Microbiota composition varies inter-individually, and effected by various environmental factors. The recent study only sampled the cats stool at one point of time. It is suggested to sample the stool similarly like urine, and sequence the stool concurrently i.e. time series analysis for the gut microbiome.

We thank the reviewer for raising this important point. For this gut microbiome analysis, we used feces collected as close to the end of the two-week experimental period as possible. Although only suggestive, this shows an association between the frequency of interactions among individuals during two weeks and the similarity of the gut microbiome as close to the end of that period as possible. 

As pointed out by the reviewer, microbiome composition varies inter-individually and is affected by various environmental factors, including food, the presence of other individuals, interaction with others, and so on. Sampling at various time points and examination of the relationship between changes of gut microbiomes in the time series and the frequency of contact with other individuals would be required. In addition, the possibility of transmission of bacteria through a shared environment has not been completely eliminated, and this is an issue that needs to be investigated as a future task. We have added these concerns to the discussion (lines 325-329).

2. The grouping strategy was vaguely described. 

Reply: As for the grouping, it was completely random. Since the number of males and females was not consistent, we have added to the discussion that it will be necessary to conduct experiments in groups with a consistent number of males and females, as well as in groups with only males or only females to understand the adaptive significance of group formation (lines 351-352).

3. While the hormonal analyses were conducted for 50 plus samples, the microbiome analysis was conducted only on 8 cats subjects, with Group 1 only represented by 1 subjects.

Reply: This was a matter of concern to us. We could not collect many feces from the cats, but this was due to the fact that we only collected feces when we could see who had defecated at the time of defecation, and the cats often defecated when humans were not around. We have added that future analyses of time series with larger sample sizes are needed (lines 325-329). 

4. Sample preparation (microbial genomics library preparation) was vaguely described.

Reply: We thank the reviewer for pointing out this point. We have added detailed information about the sample preparation, such as microbial genomics library preparation (lines 156-175).

5. Microbial genomics analysis was vaguely described.

Reply: We have added detailed information about the microbial genomics analysis (lines 176-191).

6. The raw sequencing data were not publicly available. Data availability. We recommend that all data and related meta-data underlying the findings reported in a submitted manuscript will be deposited in an appropriate public repository (GenBank, European Nucleotide Archives, National Center for Biotechnology and Information, National Bioscience Data Center, DNA data bank of Japan).

Reply: We thank the reviewer for the suggestion. The data has been deposited in the DNA Data Bank of Japan (https://www.ddbj.nig.ac.jp/index.html: accession number PRJDB12856). 

7. Is there any specific reason to opt for UniFrac distance?

Reply: We opted for the UniFrac distance because it enables us to compare the differences in the overall microbiome between two samples while considering phylogeny. The UniFrac distance takes into account the distance between bacterial phylogeny, which enables principal coordinate analysis. 

8. The manuscript was written in standard English, with minor grammatical errors.

Reply: Our manuscript was again corrected by native speakers.

9. Data presentation using PCoA were OK, but making individual comparison for microbiome analysis was vary confusing. I would suggest for the authors to conduct group comparison for the microbiome analysis. In fact, inter group correlation between behaviours also can be conducted

Reply: We thank the reviewer for the suggestion. We agree on the relevance of conducting an intergroup comparison of the gut microbiome, but this was not possible due to the insufficient number of cases (only one sample in the first group and two samples in the second group). When we show that different groups have different gut microbiome compositions , we cannot exclude the possibility of similarity from the shared environment. Therefore, we examined the relationship between the frequency of contact between individuals and the degree of similarity of their gut microbiome. To avoid any confusion, we have reorganized the results section (lines 243-250).

---

## [Decision Letter · Decision Letter 1]

2 Mar 2022

PONE-D-21-34568R1Correlation between behavior and hormone or gut microbiome implies that domestic cats (Felis silvestris catus) living in a group are not like ‘groupmates’PLOS ONE

Dear Dr. Koyasu,

Thank you for submitting your manuscript to PLOS ONE. After careful consideration, we feel that it has merit but does not fully meet PLOS ONE’s publication criteria as it currently stands. Therefore, we invite you to submit a revised version of the manuscript that addresses the points raised during the review process. Please note both reviewer #1 and #2 are concerned about the statistics used (e.g. the unbalanced sample size, how the repeated measures were accounted for). Further, the discussion and conclusion should be substantially revised to address all the comment raised. Please pay attention to the specific terms used to ensure that the sentences are more nuanced.  

We look forward to receiving your revised manuscript.

Kind regards,

Chun Wie Chong

Academic Editor

PLOS ONE

Reviewers' comments:

Reviewer's Responses to Questions

**Comments to the Author**

1. If the authors have adequately addressed your comments raised in a previous round of review and you feel that this manuscript is now acceptable for publication, you may indicate that here to bypass the “Comments to the Author” section, enter your conflict of interest statement in the “Confidential to Editor” section, and submit your "Accept" recommendation.

Reviewer #1: All comments have been addressed

Reviewer #3: All comments have been addressed

Reviewer #4: (No Response)

2. Is the manuscript technically sound, and do the data support the conclusions?

Reviewer #1: No

Reviewer #3: No

Reviewer #4: Partly

3. Has the statistical analysis been performed appropriately and rigorously? 

Reviewer #1: No

Reviewer #3: No

Reviewer #4: (No Response)

4. Have the authors made all data underlying the findings in their manuscript fully available?

Reviewer #1: Yes

Reviewer #3: Yes

Reviewer #4: Yes

5. Is the manuscript presented in an intelligible fashion and written in standard English?

Reviewer #1: No

Reviewer #3: Yes

Reviewer #4: No

6. Review Comments to the Author

Reviewer #1: I believe that the ms has improved significantly, however, I still have some doubts about the reliability of the results coming from the analysis of microbial data. The authors do not mention the use of controls and, in addition, they did not perform any filtering steps to minimise the introduction of non-real reads. Therefore, the reliability of these results is highly compromised. With regard to statistical analysis, more than one urine sample was collected from an individual to assess hormone levels, so the samples are not independent. How did you statistically analyse this dependency between the samples? Additionally, I think the authors should also modify the discussion so that the correlation between hormone levels and behaviours is easier to understand. For example: why is oxytocin in one subsection and cortisol and testosterone in another subsection? Also, at the end of the cortisol and testosterone section you mention oxytocin. So it is necessary to clarify the subsections or if they prefer to merge them all into one. It may be easier for the reader to understand the correlation of hormones and behaviour if the authors explain them all together. Since the discussion is mainly based on the correlation of hormones and affiliative contact with peers (tolerance towards other individuals), I would recommend discussing the results according to behaviour. What are the hormone levels related to affiliative behaviour? Otherwise it is very repetitive.

L28-29 Is it only the environment that shapes it? Genes have no effect? I would say that environment is one of the variables that affect behaviour. This sentence should be modified and more references added.

L33: What do you mean domesticated cats? Animals that live with humans as pets? or feral cats that live in cities. Please clarify it. I don't think cats LIVE in high densities, but rather in large groups. We can find high densities of cats in cities and towns, but not all of these cats live together as a group.

L46-47: Add References

L52. Delete this sentence, you already mentioned it in the previous sentence.

L79: did not measure the AMOUNT of specific intestinal microbiota, but rather the composition

L81-82: It is not clear to me what “solitary mammal within the same species” refers to.

L85: Indicate the name of the shelter and the locality

L86: Have these animals received any health treatment, such as antibiotics?

L113: You already mentioned how many samples you collected in L96. You only need to write it once.

L123-124: Explain why the concentrations for creatinine correction were adjusted and how it was done.

L117-125-134: There are no references on the measurement of cortisol, testosterone… Explain how these protocols were established and add references if based on previously published protocols.

L165: Add reference for primers.

L178 - Fastx tool kit needs a reference

L179: Explain the function and the package used to perform quality control. Add the corresponding references

L181:Add references for FLASH and uchimeras

L184-185: I guess you removed the chimeric sequences and then grouped them together and finally added the taxonomy. But this is not clear to me in the text, you should explain each step well and add the function used and its references.

L188-190: Just mention it in the statistical analysis section

L280: Add examples considered as affiliative behaviours in your study

L287-289: It is not clear what additional information you provide from the previous sentence

L300: Are cortisol and testosterone levels correlated? Are both related to fearlessness?

L303: Add a reference

L304-307: It is not clear which results are from this study and which are from other studies. Please clarify this and add references when referring to other studies.

Reviewer #3: Although the author has made corrections according to the recommendations given, I still see weaknesses in the study design for this study. With an unbalanced sample size for each group of cats, it is too difficult to draw conclusions. For example, the composition of the bacteria identified in this study is not discussed in depth. So it is too far -fetched for the authors to draw comparative conclusions involving the gut microbiome. Finally, I can't see the crosstalk between hormones and the composition of the cat's gut microbiota. This makes this study not strong.

Reviewer #4: Dear authors,

I appreciate the edits already completed by the authors from reviewer 1-3. However, this publication is still not at an acceptable standard for publication. The premise of the study is excellent, and it is a very interesting paper. However, authors must complete major edits, as I have begun to outline below:

The use of colloquial English ('we', "attempted to", "close to significant" etc.) must be corrected.

Abbreviations need only be explained once when first used. References must be added to statements on existing literature in the introduction, and species specific references included as opposed to blanket statements. The analytical methods used to assess hormone concentrations must be referenced. Standard units must be used (e.g. µl not ul), kits and reagents must have the manufacturing company and country of origin specified in brackets. Supplementary tables provide little extra information, where are the notes indicating what rs denotes? Why is there a ? in front of these rs scores? Consistency must be addressed e.g. "p-value" and "p". Figure's require legends which thoroughly describe the research presented. In an example, Figure 1A-C, 62 samples were analysed for hormone concentrations, yet there are 14 data points? What do the grey circle denote? Also authors must note that they are analysing the faecal microbiome, not the gastrointestinal microbiome.

Additionally, here are some specific edits I began writing:

Line 46: Please provide references for this statement

Line 59-61: Usually this is just called the gut-brain axis (GBA). References must be included at the end of this sentence.

Line 62-65: Please state which model (human, rodent, in vitro) these studies were conducted in, as there is currently no studies which have assessed this in domestic cats.

Line 67-72: Again, please state which species you are referring too. These statements form a more compelling argument for the group-living of domestic cats, if they are referring to research conducted in prides of lions as opposed to a school of dolphins.

Line 85: Please add a statement declaring where the cats were housed during the study. E.g. Adult domestic shorthair cats were housed at the XX facility/shelter, Japan for the duration of the study.

Line 89: two weeks

Line 90: litter tray?

Line 97: As diet is such an important factor for the microbiome, it would be great to have the dietary information here. Even if only to add in a sentence such as: “Cats were fed a complete and balanced commercially available kibble/can diet”

Line 113: For future studies, collect urine from individuals and note which cat produced it. This is a far more accurate way of collecting data, rather than expressing the sample as a number per individuals.

Cortisol/Creatinine/Testosterone/Oxytocin Concentrations: Were these methods derived from existing published methods? If so, please insert references.

Best wishes

7. PLOS authors have the option to publish the peer review history of their article (what does this mean?). If published, this will include your full peer review and any attached files.

Reviewer #1: No

Reviewer #3: No

Reviewer #4: No

---

## [Author Response · Author response to Decision Letter 1]

17 May 2022

Reviewer #1: 

I believe that the ms has improved significantly, however, I still have some doubts about the reliability of the results coming from the analysis of microbial data. The authors do not mention the use of controls and, in addition, they did not perform any filtering steps to minimise the introduction of non-real reads. Therefore, the reliability of these results is highly compromised. 

Reply: We appreciate the reviewer for all the detailed comments and suggestions. We found them very useful as we undertook our revision.

Instead of using the PhiX control, we have shifted the reading frame by adding 0-5 random N in front of the primer sequence. As for the non-real reads that the referee pointed out, it seems to be due to the following three issues: low quality reads, chimeras, and non-bacterial reads. We removed sequences with a quality value < 20, and all sequences were checked for chimeras, and those not determined to be chimeric were extracted and used for subsequent analyses. The information has been provided in the manuscript. For the processing of non-bacteria reads, non-bacteria reads are removed after annotation. This has been added to the methods: In addition, if there were non-bacterial reads, they were removed (lines 197-198).

With regard to statistical analysis, more than one urine sample was collected from an individual to assess hormone levels, so the samples are not independent. How did you statistically analyse this dependency between the samples? 

Reply: Thank you for pointing this out. When we checked the factors affecting all urinary hormones using a generalized linear mixed model, we found that these hormone samples were not affected by date and that individual effects were significant. Therefore, the average of the urine hormone concentrations for each individual was used as the hormone baseline for each individual. We have added an additional explanation for the urinary hormone data: The average of the urine hormone concentrations for each individual was used as the hormone baseline for each individual (lines 121-122).

Additionally, I think the authors should also modify the discussion so that the correlation between hormone levels and behaviours is easier to understand. For example: why is oxytocin in one subsection and cortisol and testosterone in another subsection? Also, at the end of the cortisol and testosterone section you mention oxytocin. So it is necessary to clarify the subsections or if they prefer to merge them all into one. It may be easier for the reader to understand the correlation of hormones and behaviour if the authors explain them all together. Since the discussion is mainly based on the correlation of hormones and affiliative contact with peers (tolerance towards other individuals), I would recommend discussing the results according to behaviour. What are the hormone levels related to affiliative behaviour? Otherwise it is very repetitive.

Reply: We appreciate the reviewer's concerns on this point. As the reviewer pointed out, the topic of oxytocin in the cortisol/testosterone section could cause confusion; therefore, the section has been reorganized. Since the study is based on the hypothesis that hormones modify cat behaviors, discussions have been made based on hormones, and we would like to maintain this approach. 

L28-29 Is it only the environment that shapes it? Genes have no effect? I would say that environment is one of the variables that affect behaviour. This sentence should be modified and more references added.

Reply: As the reviewer points out, the environment is one of the factors that affect behavior. We have corrected the relevant sentence: During evolution, ecological factors, including food availability, influence the development of group and solitary living behavior in animals [1] (lines 27-28).

L33: What do you mean domesticated cats? Animals that live with humans as pets? or feral cats that live in cities. Please clarify it. I don't think cats LIVE in high densities, but rather in large groups. We can find high densities of cats in cities and towns, but not all of these cats live together as a group.

Reply: Domesticated cat in this context refers to the domesticated cat as a species, which includes both cats living in homes with humans and feral cats. While many species of felids live alone and have exclusive territories, domestic cat species are able to live at relatively high densities in specific spaces. The definition of a group varies. Therefore, in the present context, we do not use the word 'group' but 'high density' because it is unclear what kind of group the cats form.

L46-47: Add References

Reply: We have added references.

e.g., 

M Stöwe; T Bugnyar; C Schloegl; B Heinrich; K Kotrschal; E Möstl. Corticosterone excretion patterns and affiliative behavior over development in ravens (Corvus corax). Horm Behav 53, 208–216 (2008)

FB de Waal; F Aureli; PG Judge. Coping with crowding. Sci Am 282, 76–81 (2000)

DA Gust; TP Gordon; MK Hambright; ME Wilson. Relationship between social factors and pituitary-adrenocortical activity in female rhesus monkeys (Macaca mulatta). Horm Behav 27, 318–331 (1993)

MN Barbosa; MT da S Mota. Behavioral and hormonal response of common marmosets, Callithrix jacchus, to two environmental conditions. Primates 50, 253–260 (2009)

C Crockford; RM Wittig; K Langergraber; TE Ziegler; K Zuberbuhler; T Deschner. Urinary oxytocin and social bonding in related and unrelated wild chimpanzees. Proc Biol Sci 280, 20122765 (2013)

L52. Delete this sentence, you already mentioned it in the previous sentence.

Reply: Thank you for the suggestion. The previous sentence was about various species, and the relevant sentence described cats. We have added an explanation that these describe a variety of species or cats. 

L79: did not measure the AMOUNT of specific intestinal microbiota, but rather the composition

Reply: The reviewer’s comment is correct. We have corrected the sentence: 3. there is a relationship between the composition of gut microbiomes and hormone concentrations in an individual’s body (lines 80-81)

L81-82: It is not clear to me what “solitary mammal within the same species” refers to.

Reply: ‘Solitary mammal within the same species’ means that ‘role of oxytocin in conspecific social behaviors in a solitary mammal.’ To clarify, we have edited the sentence (lines 83).

L85: Indicate the name of the shelter and the locality

Reply: We have added the name of the shelter and the locality (line 87). 

L86: Have these animals received any health treatment, such as antibiotics?

Reply: The cats in the experiment were not treated with antibiotics or other treatments.

L113: You already mentioned how many samples you collected in L96. You only need to write it once.

Reply: Thank you for pointing this out. We deleted the sentence with the same information in the experimental setting. 

L123-124: Explain why the concentrations for creatinine correction were adjusted and how it was done.

Reply: All urinary hormone concentrations vary greatly, depending on the concentration of urine. Since the daily excretion of creatinine is constant (Shaffer PA, 1908), the hormone concentration considering the concentration of urine can be calculated based on the amount of creatinine in that urine (Munro CJ et al., 1991).

Shaffer PA., The excretion of creatin and creatinine in health and disease., American Journal of Physiology, 1908;10:1-10.

Munro CJ, Stabenfeldt GH, Cragun JR, Addiego LA, Overstreet JW, Lasley BL., Relationship of serum estradiol and progesterone concentrations to the excretion profiles of their major urinary metabolites as measured by enzyme immunoassay and radioimmunoassay., Clinical Chemistry, 1991;37:838-844.

L117-125-134: There are no references on the measurement of cortisol, testosterone… Explain how these protocols were established and add references if based on previously published protocols.

Reply: These protocols for hormone assays were based on previous studies. Related references have been added.

L165: Add reference for primers.

Reply: We have added an appropriate reference for the primers. 

Klindworth, A., Pruesse, E., Schweer, T., Peplies, J., Quast, C., Horn, M., & Glöckner, F. O. (2013). Evaluation of general 16S ribosomal RNA gene PCR primers for classical and next-generation sequencing-based diversity studies. Nucleic acids research, 41(1), e1-e1.

L178 - Fastx tool kit needs a reference

Reply: We have added an appropriate reference.

A Gordon; GJ Hannon; Others. Fastx-toolkit. FASTQ/A short-reads preprocessing tools (unpublished) http://hannonlab cshl edu/fastx_toolkit 5 (2010)

L179: Explain the function and the package used to perform quality control. Add the corresponding references.

Reply: Fragment Analyzer and dsDNA 915 Reagent Kit (Advanced Analytical Technologies, Inc.) were used to check the quality of the libraries. We have added a relevant sentence and reference: The quality of libraries was evaluated using a Fragment Analyzer and a dsDNA 915 Reagent Kit (Advanced Analytical Technologies, Inc., USA).

N Susai; T Kuroita; K Kuronuma; T Yoshioka. Analysis of the gut microbiome to validate a mouse model of pellagra. Bioscience of Microbiota, Food and Health advpub, 2021–2059 (2022)

L181:Add references for FLASH and uchimeras

Reply: We have added the relevant references. 

FLASH: T Magoč; SL Salzberg. FLASH: fast length adjustment of short reads to improve genome assemblies. Bioinformatics 27, 2957–2963 (2011)

UCHIME: RC Edgar; BJ Haas; JC Clemente; C Quince; R Knight. UCHIME improves sensitivity and speed of chimera detection. Bioinformatics 27, 2194–2200 (2011)

L184-185: I guess you removed the chimeric sequences and then grouped them together and finally added the taxonomy. But this is not clear to me in the text, you should explain each step well and add the function used and its references.

Reply: We have changed the sentences: The sequences that passed all the filtering were checked for chimeras using the uchime algorithm of usearch[61]. The reads that were not determined to be chimeras were used in subsequent analyses. In addition, if there were non-bacterial reads, they were removed. The creation of OTUs and phylogenetic estimation were performed using QIIME workflow scripts with no reference and all parameters set to default conditions. The number of reads used in the QIIME analysis was 224,878 (28,110 ± 1764) (lines 195-200).

L188-190: Just mention it in the statistical analysis section

Reply: We have removed the sentence and moved it to the statistical analysis section.

L280: Add examples considered as affiliative behaviours in your study

Reply: We considered allo-grooming, entering bed and sharing bed as affiliative behaviors in the present study. Since the affiliative behavior in the previous studies is mainly allo-grooming, we have described it. 

L287-289: It is not clear what additional information you provide from the previous sentence

Reply: We have revised this part for enhanced clarity: The experimental period of two weeks may be too short to enable the formation of a tight relationship group and observing them for a longer period could result in a change in their relationships. Although their relationships were possibly established since the cats who participated in the experiment had been living in the same room at the shelter before the experiment, investigations over a longer duration could offer greater insights on group-formation in cats (lines 290-294). 

L300: Are cortisol and testosterone levels correlated? Are both related to fearlessness?

Reply: Yes, there was a positive correlation between cortisol and testosterone concentrations. Yes, both are related to fearlessness.

L303: Add a reference

Reply: This is a suggestion based on the results of the present study and there is no previous study reporting such results. A comparison should be made to examine whether testosterone is associated with any of the behaviors in castrated and uncastrated cats. To clarify, we have clearly stated that this sentence is derived from the results of the present study. 

L304-307: It is not clear which results are from this study and which are from other studies. Please clarify this and add references when referring to other studies.

Reply: Same as in the previous point, we have made them clearer.

 

Reviewer #3: 

Although the author has made corrections according to the recommendations given, I still see weaknesses in the study design for this study. With an unbalanced sample size for each group of cats, it is too difficult to draw conclusions. For example, the composition of the bacteria identified in this study is not discussed in depth. So it is too far -fetched for the authors to draw comparative conclusions involving the gut microbiome. Finally, I can't see the crosstalk between hormones and the composition of the cat's gut microbiota. This makes this study not strong.

Reply: We appreciate you pointing this out. The fecal sample size in this study is a a major source of concern. Although it would have been ideal to collect feces from all individuals, limited feces were collected from all individuals within the two weeks. Some of the cats were afraid of people and would not defecate when people were present, because they were from a shelter. Therefore, data are limited due to the small sample size, and the results of the present study only suggest that there is a relationship between gut microbiome and behaviors and hormones. Nevertheless, the preliminary results could guide and facilitate future studies on cat microbiomes and the correlation with behavior. 

 

Reviewer #4: 

Dear authors,

I appreciate the edits already completed by the authors from reviewer 1-3. However, this publication is still not at an acceptable standard for publication. The premise of the study is excellent, and it is a very interesting paper. However, authors must complete major edits, as I have begun to outline below:

Reply: Thank you very much for providing such critical comments. In the following sections, you will find our responses to each of your points and suggestions. We are grateful for the time and energy you expended in reviewing this manuscript.

The use of colloquial English ('we', "attempted to", "close to significant" etc.) must be corrected. Abbreviations need only be explained once when first used. 

Reply: We have revised the entire manuscript accordingly, with checks by native speakers.

References must be added to statements on existing literature in the introduction, and species specific references included as opposed to blanket statements. 

Reply: We have added species-specific references (cats/other felids) and noted the associated species.

M Franchini; A Prandi; S Filacorda; EN Pezzin; Y Fanin; A Comin. Cortisol in hair: a comparison between wild and feral cats in the north-eastern Alps. Eur J Wildl Res 65, 90 (2019)

H Finkler; J Terkel. Cortisol levels and aggression in neutered and intact free-roaming female cats living in urban social groups. Physiol Behav 99, 343–347 (2010)

G Genaro; CR Franci. Cortisol influence on testicular testosterone secretion in domestic cat: An in vitro study. Pesqui Vet Bras 30, 887–890 (2010)

The analytical methods used to assess hormone concentrations must be referenced. Standard units must be used (e.g. µl not ul), kits and reagents must have the manufacturing company and country of origin specified in brackets. 

Reply: We have added the appropriate references. In addition, standard units have been corrected and countries of origin for kits and reagents added.

Supplementary tables provide little extra information, where are the notes indicating what rs denotes? Why is there a ? in front of these rs scores? Consistency must be addressed e.g. "p-value" and "p". Figure's require legends which thoroughly describe the research presented. In an example, Figure 1A-C, 62 samples were analysed for hormone concentrations, yet there are 14 data points? What do the grey circle denote? 

Reply: Thank you for pointing this out. The ‘rs’ represent correlation coefficients. The text in the supplemental table was garbled and a ‘?’ showed a negative value. We have corrected these points. In addition, explanations have been added to describe the result presented comprehensively.

Also authors must note that they are analysing the faecal microbiome, not the gastrointestinal microbiome.

Reply: Thank you for pointing this out. We have added the sentence: Most mammalian gut microbiome studies have focused on fecal microbiota, but it is unclear how well fecal samples reflect the microbiota of the intestinal region, and these should also be examined (lines 352-354).

Additionally, here are some specific edits I began writing:

Line 46: Please provide references for this statement

Reply: We have added the relevant references.　

e.g., 

M Stöwe; T Bugnyar; C Schloegl; B Heinrich; K Kotrschal; E Möstl. Corticosterone excretion patterns and affiliative behavior over development in ravens (Corvus corax). Horm Behav 53, 208–216 (2008)

FB de Waal; F Aureli; PG Judge. Coping with crowding. Sci Am 282, 76–81 (2000)

DA Gust; TP Gordon; MK Hambright; ME Wilson. Relationship between social factors and pituitary-adrenocortical activity in female rhesus monkeys (Macaca mulatta). Horm Behav 27, 318–331 (1993)

MN Barbosa; MT da S Mota. Behavioral and hormonal response of common marmosets, Callithrix jacchus, to two environmental conditions. Primates 50, 253–260 (2009)

C Crockford; RM Wittig; K Langergraber; TE Ziegler; K Zuberbuhler; T Deschner. Urinary oxytocin and social bonding in related and unrelated wild chimpanzees. Proc Biol Sci 280, 20122765 (2013)

Line 59-61: Usually this is just called the gut-brain axis (GBA). References must be included at the end of this sentence.

Reply: We have added the relevant references.

JF Cryan; KJ O’Riordan; CSM Cowan; KV Sandhu; TFS Bastiaanssen; M Boehme; MG Codagnone; S Cussotto; C Fulling; AV Golubeva; KE Guzzetta; M Jaggar; CM Long-Smith; JM Lyte; JA Martin; A Molinero-Perez; G Moloney; E Morelli; E Morillas; R O’Connor; JS Cruz-Pereira; VL Peterson; K Rea; NL Ritz; E Sherwin; S Spichak; EM Teichman; M van de Wouw; AP Ventura-Silva; SE Wallace-Fitzsimons; N Hyland; G Clarke; TG Dinan. The Microbiota-Gut-Brain Axis. Physiol Rev 99, 1877–2013 (2019)

Line 62-65: Please state which model (human, rodent, in vitro) these studies were conducted in, as there is currently no studies which have assessed this in domestic cats.

Reply: Thank you for the suggestion. For enhanced clarity, we have added the models in the studies. 

Line 67-72: Again, please state which species you are referring too. These statements form a more compelling argument for the group-living of domestic cats, if they are referring to research conducted in prides of lions as opposed to a school of dolphins.

Reply: These references were mainly about insects and rodents. We have identified the species examined in the studies.

Line 85: Please add a statement declaring where the cats were housed during the study. E.g. Adult domestic shorthair cats were housed at the XX facility/shelter, Japan for the duration of the study.

Reply: All cats were housed in one room at Azabu University during the experiment. We have added information on where the animals were maintained during the experiment: All cats were housed in one room at Azabu University during the experiment.(lines 89-90).

Line 89: two weeks

Reply: We have corrected (line 93).

Line 90: litter tray?

Reply: Yes, those were litter trays. We have changed ‘toilets’ to ‘litter trays (line 94).’ 

Line 97: As diet is such an important factor for the microbiome, it would be great to have the dietary information here. Even if only to add in a sentence such as: “Cats were fed a complete and balanced commercially available kibble/can diet”

Reply: We have added a sentence about diet: All cats were fed a complete and balanced commercially available kibble diet (lines 100-101). 

Line 113: For future studies, collect urine from individuals and note which cat produced it. This is a far more accurate way of collecting data, rather than expressing the sample as a number per individuals.

Reply: Thank you for your comment. In the present study, we recorded which cats excreted and the time. A minimum 1 - maximum 9 urine samples were collected per individual. We have added the sentence in the hormonal assay section: Sixty-three urine samples, i.e., 4.2 ± 2.4 samples/individual (A minimum of one and a maximum of nine urine samples were collected per individual) were collected during the entire observation period… (lines 116-117). 

Cortisol/Creatinine/Testosterone/Oxytocin Concentrations: Were these methods derived from existing published methods? If so, please insert references.

Reply: We have added the relevant references.

T Nagasawa; M Ohta; H Uchiyama. The Urinary Hormonal State of Cats Associated With Social Interaction With Humans. Front Vet Sci 8, 680843 (2021)

K Uetake; A Goto; R Koyama; R Kikuchi; T Tanaka. Effects of single caging and cage size on behavior and stress level of domestic neutered cats housed in an animal shelter. Anim Sci J 84, 272–274 (2013)

K Carlstead; JL Brown; W Strawn. Behavioral and physiological correlates of stress in laboratory cats. Appl Anim Behav Sci 38, 143–158 (1993)

K Kojima; S Hguchi; N Tashiro. Changes in urinary cortisol associated with hypthalamically elicited restlessness. Stress Med 11, 61–65 (1995)

---

## [Editor Report · Decision Letter 2]

25 May 2022

Correlations between behavior and hormone concentrations or gut microbiome imply that domestic cats (Felis silvestris catus) living in a group are not like ‘groupmates’

PONE-D-21-34568R2

Dear Dr. Koyasu,

We’re pleased to inform you that your manuscript has been judged scientifically suitable for publication and will be formally accepted for publication once it meets all outstanding technical requirements.

Kind regards,

Chun Wie Chong

Academic Editor

PLOS ONE
---

## [Editor Report · Acceptance letter]

30 May 2022

PONE-D-21-34568R2 

Correlations between behavior and hormone concentrations or gut microbiome imply that domestic cats (*Felis silvestris catus*) living in a group are not like ‘groupmates’ 

Dear Dr. Koyasu:

I'm pleased to inform you that your manuscript has been deemed suitable for publication in PLOS ONE. Congratulations! Your manuscript is now with our production department. 

Kind regards, 

on behalf of

Dr. Chun Wie Chong 

Academic Editor

PLOS ONE